# Changes in anthropometric indices, lifestyle patterns, and mental stress with ramadan intermittent fasting among healthy students: A prospective cohort study

Anfal AL-Dalaeen[1]*, Eman Alhasan[1], Eman Saleh[1], Sally Atawneh[2], Tala Barakat[1]

**1** Department of Clinical Nutrition and Dietetics, Faculty of Allied Medical Sciences, Applied Science Private University, Amman, Jordan, **2** Department of Clinical Pharmacy and Therapeutics, Applied Science Private University, Amman, Jordan

* a_dalaeen@asu.edu.jo

## Abstract

### Background

The association between Ramadan Intermittent Fasting (RIF) and metabolic diseases has been studied. However, there remains a significant gap in the research regarding the changes in anthropometric indices related to body weight during RIF, as these have not been extensively investigated. Additionally, the role of lifestyle behaviours in relation to RIF has not been thoroughly studied. This paper aims to address these gaps by examining the effect of RIF on anthropometric indices, lifestyle patterns, and mental stress among university students.

### Methods

A prospective cohort study included 150 university students aged 18–40 years. Data was collected using questionnaires and face-to-face interviews conducted one week before and immediately after one week of RIF. Body weight and height were measured to calculate body mass index (BMI) and other anthropometric indices related to obesity, including Conicity Index (CI), Abdominal Volume Index (AVI), Body Adiposity Index (BAI), Weight-Adjusted-Waist Index (WWI), and Body Roundness Index (BRI).

### Results

The Waist-to-Height Ratio (WHtR) showed a significant decrease from $0.481 \pm 0.11$ to $0.45 \pm 0.10$, with reductions in body weight (kg) ($62.63 \pm 1.2$ vs. $69.25 \pm 1.18$, $p < 0.05$). A significant change in BRI before and after fasting ($1.84 \pm 0.37$ vs. $1.06 \pm 0.17$, $p < 0.05$) was observed. However, there was no significant difference in other anthropometric indices such as CI, WWI, and BAI. Significant changes were observed in lifestyle parameters after RIF, including decreased sleep hours ($6 \pm 0.263$ vs.

**Data availability statement:** All relevant data are within the manuscript.

**Funding:** The author(s) received no specific funding for this work.

**Competing interests:** The authors have declared that no competing interests exist.

$8 \pm 0.581$, $p < 0.05$) and reduced number of meals after fasting ($1 \pm 0.135$ vs. $3 \pm 1.242$, $p < 0.05$). Stress levels significantly decreased after RIF ($p < 0.05$). RIF fasting positively impacts body weight, anthropometric indices, and stress levels among healthy students.

## Conclusion

These findings suggest that RIF can improve health outcomes by promoting better dietary habits, enhancing sleep duration, and reducing stress, leading to significant reductions in body weight, BMI, and BRI.

---

## 1. Introduction

Obesity is a chronic disease that is increasing worldwide, and now it is considered a global epidemic [1]. It is defined as an abnormal accumulation of fat in the body, which negatively affects health [2]. Traditionally, obesity is assessed using body mass index (BMI), waist circumference (WC), hip circumference (HC), and waist-to-height ratio (WHtR). However, alternative assessment methods scales such as the Conicity index (CI), Body adiposity index (BAI), Weight-adjusted-waist index (WWI), and A body shape index (ABSI) offer a more accurate assessment of visceral fat accumulation [3]. These parameters provide better predictions for the onset of metabolic syndrome (MetS) and cardiovascular risk. [4].

Major factors of a healthy lifestyle include eating a healthy nutritious diet, increasing physical activity, maintaining normal body weight, and adequate sleep hours. Modifying these factors plays a fundamental role in promoting physical and mental well-being and preventing chronic illnesses [5]. There are various methods for weight loss, including physical activity, specific diets, medication, and hormonal treatments. Among these, intermittent fasting has gained attention for its effectiveness in weight loss [6]. Research studies have shown that the weight loss achieved through intermittent fasting leads to various benefits and reduces the risks of related conditions [7,8]. Fasting is considered a healthy practice, with recent studies showing a positive association between various forms of fasting, including intermittent fasting and Ramadan fasting, and longevity [9]. Ramadan, the holy month during which Muslims fast daily from dawn to sunset [10]. It is a form of intermittent fasting that is safe for all healthy individuals. It can lead to significant changes in body weight and health improvement. Studies have also found intermittent fasting during Ramadan is associated with decreasing stress and anxiety [6]. However, changes in meal timing and lifestyle habits after Ramadan Intermittent Fasting (RIF) may affect circadian rhythm. leading to changes in sleep and stress levels [11].

Even though intermittent fasting is associated with weight loss, research on the impact of fasting after RIF on weight has produced conflicting results [12]. Some studies indicate no change in weight or even weight gain, while others suggest weight loss [13]. For instance, studies among Pakistani individuals have shown that dietary habits, physical activity levels, food preferences, and sleep patterns during

RIF differ significantly from those in other countries, potentially leading to weight gain rather than weight loss [14,15]. Similar patterns have been observed in Saudi Arabia [16]. Additionally, there is a lack of studies addressing the association between RIF and anthropometric indices, mental stress, and lifestyle habits. Therefore, we are conducting this study to comprehensively investigate the effects of RIF on indices of obesity, lifestyle, and mental stress, aiming to fill these critical gaps in the literature and provide clearer insights into these associations.

## 2. Materials and methods

### 2.1. Study design, area, and period

This prospective cohort study was conducted during the months of Shaban and Ramadan (March to April 2024) to evaluate the impact of RIF as a dependent variable on anthropometric indices, dietary habits, physical activity, and mental stress among a group of university students aged between 18 and 40 years at Applied Science Private University, Amman, Jordan. The participants resided in university housing and private accommodations. The fasting period lasted from 23 to 25 days for premenopausal female participants, whereas for male, it lasted from 28 to 30 days.. During this period, the average temperature in Amman ranged from 14°C to 25°C, and the exact duration of fasting each day was approximately 13–14 hours. Since metabolic changes induced by RIF are transitory and revert to pre-fasting levels after one month of fasting cessation, it is important to assess these changes after the fasting period to understand their immediate impact [17]. Before the study began, written informed consent was acquired from each subject. The Applied Science Private University Ethics Committee authorized the protocols, tools, and procedures for this project (2023-PHA-20).

### 2.2. Study participants and sampling procedure

A prospective cohort study was conducted on 150 university students, aged 18–40 years old. Participants were recruited using a convenience sampling technique among eligible university students who intended to fast during Ramadan. The inclusion criteria were healthy students who did not complain of any chronic diseases or mental illnesses and were willing to fast during Ramadan 2024. A calculation of the sample size was conducted to ascertain the necessary number of participants. Drawing from earlier quasi-experimental studies, it was determined that at least 70 participants were required to attain 80% power with a p-value of 0.05 in order to identify a significant difference of 1 kg in BMI before and after fasting. This estimation was based on previous research indicating an anticipated effect size of 0.20 [18]. The data were collected before one week of Ramadan and immediately after one week of RIF, as in Fig 1.

### 2.3. Data collection

Using standardized questionnaires and face-to-face interviews conducted by trained and qualified personnel, including dietitians, demographic data, including age, sex, marital status, education level, lifestyle data (physical activity and sleep hours), and medical history were gathered. The interviews were conducted in person to ensure consistency and accuracy in data collection. Dietary habits were assessed by a trained dietician at two points: the initial visit and the follow-up immediately after one week of RIF. Information on dietary intake was obtained using a validated short food frequency questionnaire containing 22 items to evaluate fat, sugar, fruit, vegetable, and water intake [19].

The Perceived Stress Scale (PSS) was used to assess stress. The PSS-10 questionnaire has undergone validation for assessing perceived stress in university students [20]. The scoring for the PSS ranges from 0 to 40, with scores of 0–13 indicating low stress, scores between 14 and 26 representing moderate stress, and scores from 27 to 40 signifying high levels of perceived stress. PSS scores were taken for the month before RIF and after one week of RIF.

Physical activity was assessed as Metabolic equivalents (METs) using the Seven-Day Physical Activity Recall validated questionnaire [21]. The participants were asked to recall morning, afternoon, and evening activities on the previous day, for a full seven days of information on minutes spent engaged in vigorous-intensity and moderate-intensity activities.

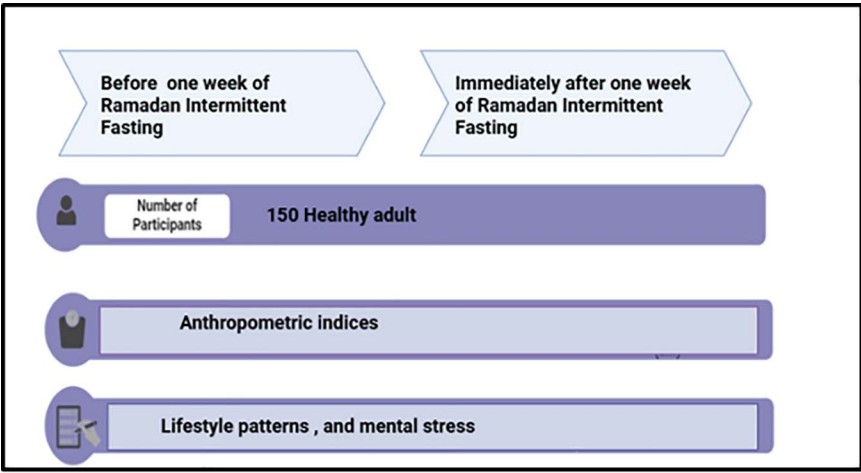

**Fig 1. Overview of study design.**

To ensure that changes in outcomes were truly attributable to Ramadan fasting, baseline measurements of dietary habits, physical activity levels, and mental health status were taken before the study began. These control variables were considered in the analysis to isolate the effects of RIF on the observed changes in anthropometric indices, lifestyle patterns, and mental stress.

**2.3.1. Anthropometric indices.** Using a standardized protocol, well-trained examiners measured anthropometric parameters. The participants were asked to wear minimal clothing [22]. Body weight was measured using a weight scale to the nearest 0.1 kg, while the height was measured by the stadiometer (BSM 170˚) device, USA., WC and HC were measured while standing using the RIEDER Body Measure 60/150 cm (Inct. Bonus Kit, REIDHK®, China). WC was measured on the horizontal plane midway between the lowest rib and the iliac crest. HC was measured by wrapping the measuring tape around the widest part of the hips, ensuring it was parallel to the floor. BMI and WHtR were calculated based on these measurements.

Anthropometric indices related to obesity include the body adiposity index (BAI), the body shape index (ABSI), the Body Roundness Index (BRI), and the Weight-adjusted weight index (WWI). The following are the lists of mathematical formulas that were used to calculate these anthropometric indices [23–27]:

$$A \text{ body shape index (ABSI)} = WC \text{ (cm)}/(BMI^{2/3} \text{ Height (cm)}^{1/2})$$

$$Weight-adjusted \text{ weight index (WWI)} = (Waist \text{ Circumference (cm)})/Weight \text{ (Kg)})^{1/2}$$

$$Body \text{ roundness index (BRI)} = 364.2 - 365.5 \sqrt{1 - \left( \frac{(Waist\ circumference(cm)/2\pi)^2}{(0.5*Height\ (cm))^2} \right)}$$

$$Body \text{ Adiposity Index (BAI)} = \frac{Hip \text{ circumference (cm)}}{(Height\ (m))\ 1.5} - 18$$

$$Conicity \text{ Index (CI)} = Waist \text{ circumference (m)}/(0.109 * \sqrt{\left( \frac{Weight(kg)}{height(m)} \right)})$$

## 2.4. Ethical consideration

The Institutional Review Boards at Applied Science Private University approved all recruitment and consent procedures and informed consent that explained all the duties and rights of the participant were signed by all participants.

## 2.5. Statistical analysis

Statistical analysis was conducted on categorical variables, presented as frequencies and percentages, while continuous variables were presented as mean±SD. A paired sample t-test analysis was employed to compare data between the pre- and post-RIF assessments. Significance was determined at a p-value of ≤ 0.05. Additionally, Cohen's d effect sizes, 95% confidence intervals (95% CI) were calculated to provide a sense of the practical significance of the findings, aiding in understanding the magnitude of the changes observed. The Statistical Package for Social Sciences version 23.0 software (SPSS Inc., Chicago, IL, USA) was utilized for all data analysis purposes.

## 3. Results

A total of 150 (100 female and 50 male) participated in this study. With 50.6% of the age range of 19–24 years, most of the respondents were single (76.1%), bachelor's Students (79.5%), non-working (76%), with incomes less than 352 US$ (37.5%), as shown in Table 1.

Table 2 provides anthropometric and obesity indices before and after RIF. The mean weight was 67.25 kg, before RIF, which decreased to 63.63±21.18 kg after RIF (p<0.05). Also, A significant decrease in the BMI of participants after RIF as compared to before RIF (26.085±5.07, 23.048±5.15, p=0.030, Cohen's d=0.59). The Waist-to-Height Ratio (WHtR) Ratio showed a significant decrease from 0.481 to 0.45 (p=0.011, Cohen's d=0.29). Significantly decreased from 1.84 to 1.06 (p=0.001, Cohen's d=0.72). NO significant changes were observed in CI, WWI, BAI, and ABSI (p>0.05). BRI: The

**Table 1. Sociodemographic Characteristics of Study Participants.**

| Variable | | n | % | P-value |
|---|---|---|---|---|
| **Sex** | Male | 58 | 36.7 | <0.01 |
| | Female | 92 | 63.3 | |
| **Age (years)** | 19-24 | 85 | 58.7 | <0.01 |
| | 25-30 | 45 | 25.9 | |
| | 30-40 | 20 | 15.4 | |
| **Social Status** | Single | 110 | 76.1 | <0.01 |
| | Married | 30 | 22.9 | |
| | Divorced | 10 | 1 | |
| **Educational level** | High school | 10 | 3.1 | <0.01 |
| | Diploma | 20 | 7.4 | |
| | Bachelor's | 100 | 79.5 | |
| | Postgraduate | 20 | 10 | |
| **Working Status** | Working | 30 | 24 | <0.01 |
| | Not working | 120 | 76 | |
| **Income (Dolar)** | lower than 352 | 60 | 37.5 | >0.05 |
| | 353-700 | 37 | 24 | |
| | 700-1000 | 15 | 11.6 | |
| | 1000-1400 | 18 | 13.1 | |
| | more than 1500 | 20 | 14 | |

**Table 2. Different in Anthropometric indices Before and Immediately After Ramadan intermittent fasting among Study participant (*n*=150).**

| Variables | | Mean | P- value | Cohen's d |
|---|---|---|---|---|
| **Weight (kg)** | Before | 67,25±1.18 | 0.02 | 0.31 |
| | After | 63.63±1.18 | | |
| **Height (cm)** | Before | 164.63±8.72 | 0.103 | 0.21 |
| | After | 162.86±6.75 | | |
| **BMI (kg/m²)** | Before | 26.08±5.07 | 0.030 | 0.59 |
| | After | 23.05±5.15 | | |
| **Waist Circumference (cm)** | Before | 76.34±21.57 | 0.11 | 0.20 |
| | After | 72.25±18.70 | | |
| **WHtR** | Before | 0.481±0.106 | 0.011 | 0.29 |
| | After | 0.45±0.104 | | |
| **ABSI** | Before | 0.082±0.014 | 0.304 | 0.20 |
| | After | 0.071±0.02 | | |
| **CI** | Before | 1.32±0.58 | 0.82 | 0.04 |
| | After | 1.30±0.39 | | |
| **WWI** | Before | 1.06±0.16 | 0.743 | 0.26 |
| | After | 1.02±0.14 | | |
| **BRI** | Before | 1.84±0.37 | 0.001 | 0.72 |
| | After | 1.06±0.17 | | |
| **BAI** | Before | 1.99±0.24 | 0.611 | 0.30 |
| | After | 1.84±0.10 | | |

BMI: Body Mass index; WHtR: Waist-to-height ratio; CI Conicity Index; BAI: Body adiposity index; BRI: Body Roundness Index; WWI: Weight-adjusted-waist index ABSI: A body shape index. P value was completed using Paired-Samples T Test to compare the results of Before- and during Ramadan.

Body Roundness Index significantly decreased from 1.84 to 1.06 (p=0.001, Cohen's d=0.72), indicating a large effect size. This highlights substantial changes in body shape and fat distribution.

Other Indices (CI, WWI, BAI, ABSI): No significant changes were observed in CI, WWI, BAI, and ABSI (p>0.05). The effect sizes for these indices were small, suggesting that these measures are less sensitive to short-term changes in body composition during Ramadan fasting. After Ramadan, A significant change was found in sleep hours before and after Ramadan (8±0.581, 6±0.263, p=0.03, Cohen's d=0.42). Moreover, the number of meals per day significantly decreased from 3 to 1 (p=0.02, Cohen's d=0.58). Conversely, there was a significant increase in the percentage of participants with low stress (from 13% to 19%, p=0.03, Cohen's d=0.18). There was no significant difference in the sum of MetS before and after the RIF (p=0.0645, Cohen's d=0.35), as shown in Table 3.

The study examined dietary consumption patterns before and after RIF among participants, as shown in Table 4. The consumption of vegetables increased significantly after RIF as compared to before RIF (p<0.05). However, fruit consumption after RIF decreased. However, there were no changes in the use of artificial sweeteners or caffeine consumption after RIF as compared to before RIF (p>0.05).

## 4. Discussion

The RIF was associated with reductions in body weight, BMI, WHtR, and BRI among healthy university students. In contrast, other anthropometric indices such as CI, WWI, BAI, and ABSI did not show significant changes. In addition to anthropometric modifications, RIF was correlated with lifestyle changes, including reduced sleep duration and decreased meal frequency. Notably, perceived stress levels improved following the fasting period. These findings suggest that RIF may induce measurable physiological and behavioral adaptations among healthy adults.

**Table 3. Different in lifestyle before and immediately after Ramadan intermittent fasting.**

| Variable | | Before- Ramadan | After- Ramadan | *P* value | Cohen's d |
|---|---|---|---|---|---|
| Sleep hours, (mean±SD) | | 8±0.58 | 6±0.26 | 0.03 | 0.42 |
| Physical activity (MetS), (mean±SD) | | 2186.43±23.40 | 2023.36±22.79 | 0.0645 | 0.35 |
| Number of meals/24 hr, (mean±SD) | | 3±1.24 | 1±0.13 | 0.02 | 0.58 |
| Perceived Stress level (*n*, %) | Low | 11 (13%) | 15 (19%) | | |
| | Moderate | 114 (56%) | 125 (60%) | | 0.18 |
| | High | 25 (21%) | 20 (21%) | | |

P value was completed using Paired-Samples T Test to compare the results of pre- and during Ramadan. MET; metabolic equivalents.

The association between body weight and fasting during Ramadan is well-documented. In the current study, subjects experienced a significant weight reduction, averaging 5 kg. Systematic reviews and meta-analyses support these findings, showing mean weight losses of 1.51 kg for men and 0.92 kg for women [28]. Another review highlights significant weight loss in males but not in females, suggesting gender-specific physiological responses to fasting [29]. Additionally, greater weight loss was observed in participants with higher pre-Ramadan BMI, indicating that initial body composition influences weight loss outcomes [29].

The mechanism involves reduced caloric intake, leading to a negative energy balance, prompting the body to utilize stored fat for energy. Hormonal changes, such as decreased insulin levels and increased glucagon, facilitate lipolysis and fat oxidation [30]. Dehydration could also contribute to weight loss during fasting, as the body loses water through sweating, breathing, and metabolic functions [31], which is proven by our result of total water drinking $p < 0.001$). Post-Ramadan, weight regain is common due to increased caloric intake and reduced physical activity [32]. Understanding these patterns is crucial for designing effective weight management strategies during and after Ramadan.

The reduction in BMI is significant, but not with other anthropometric indices related to obesity. This is likely due to the significant increase in consumption of pastries, desserts, and processed meat, which affects most anthropometric indices (i.e., BMI, WC, ABSI, CI, and WWI).

The current anthropometric measurement results are consistent with findings from Alzoughool et al. (2019) study among healthy students after RIF [33]. Also, Khan et al. (2017) found that there was no significant change in the anthropometric parameters of the thirty-five medical students in Pakistan; these parameters include BMI, WHtR, BAI, and VAI [15]. Controversially, there are a tremendous number of studies that indicate RIF may be considered an effective weight-loss technique. A recent LORANS study and meta-analysis by Al-Jafar et al. (2023) showed that there is a significant decrease in all anthropometric parameters, including weight, WC, BMI, HC, and WHR, in addition to fat mass and fat-free mass [34].

The change in WC was addressed partially as a result of reduced total body water and fat mass. Other recent studies also showed the same results regarding the effect of RIF on weight loss and other anthropometric [32]. On the other hand, Das et al. (2019) conducted a study on a population of Indian Muslims after Ramadan. The results showed an increase in body weight and BMI after one month of Ramadan [35]. Additionally, there is a lack of studies examining the effects of RIF fasting on Conicity Index (CI), Abdominal Volume Index (AVI), and Body Adiposity Index (BAI). The current study found that there was no significant difference among these parameters. The mechanism behind the lack of significant differences in CI, AVI, and BAI during RIF could be attributed to the body's metabolic adaptations [7]. During fasting, the body shifts from using glycogen stores to burning fat for energy, leading to overall weight and fat mass reduction [32]. However, these changes may not significantly impact specific adiposity indices like CI, AVI, and BAI, which are more sensitive to long-term body composition changes rather than short-term weight fluctuations.

**Table 4. Changes in Dietary Consumption Patterns Before and Immediately After Ramadan intermittent fasting.**

| | Dietary Patterns | Before- Ramadan | Before- Ramadan | P Value |
|---|---|---|---|---|
| | | n | n | |
| Vegetables (Serving/day)/ day | 0 | 10 | 26 | 0.04 |
| | 1-2 | 17 | 24 | |
| | >3 | 0 | 5 | |
| Fruits (Serving/day) | 0 | 27 | 13 | |
| | 1 | 44 | 16 | 0.01 |
| | 2 | 23 | 14 | |
| | 3 | 5 | 14 | |
| | 4 | 0 | 14 | |
| | 5 | 0 | 13 | |
| | 6 | 0 | 14 | |
| Fries/day Surving you used | 0 | 15 | 10 | |
| | 1 | 26 | 13 | 0.030 |
| | 2 | 18 | 35 | |
| | 3 | 20 | 19 | |
| | 4 | 11 | 13 | |
| | 5 | 3 | 6 | |
| | 6 | 2 | 2 | |
| | 7 | 4 | 1 | |
| Red meat/day | 0 | 10 | 47 | |
| | 1 | 32 | 27 | 0.001 |
| | 2 | 21 | 10 | |
| | 3 | 21 | 9 | |
| | 4 | 9 | 3 | |
| | >5 | 5 | 2 | |
| Processed Meat/day | 0 | 43 | 14 | |
| | 1 | 23 | 16 | 0.001 |
| | 2 | 17 | 20 | |
| | 3 | 8 | 29 | |
| | 4 | 5 | 9 | |
| | >5 | 3 | 11 | |
| Desserts/day | 0 | 6 | 0 | |
| | 1 | 21 | 2 | 0.001 |
| | 2 | 18 | 8 | |
| | 3 | 16 | 13 | |
| | 4 | 8 | 22 | |
| | 5 | 13 | 11 | |
| | 6 | 5 | 9 | |
| | >7 | 12 | 34 | |
| Water (cup) /Day | 0 | 13 | 14 | |
| | 1 | 5 | 14 | 0.001 |
| | 2 | 5 | 14 | |
| | 3 | 12 | 14 | |
| | 4 | 10 | 8 | |
| | 5 | 12 | 17 | |
| | 6 | 15 | 4 | |
| | >7 | 27 | 14 | |

*(Continued)*

**Table 4.** (Continued)

| | Dietary Patterns | Before- Ramadan | Before- Ramadan | P Value |
|---|---|---|---|---|
| | | n | n | |
| Juice (cup)/Day | 0 | 35 | 12 | 0.300 |
| | 1 | 19 | 22 | |
| | 2 | 18 | 30 | |
| | 3 | 4 | 16 | |
| | 4 | 5 | 10 | |
| | >5 | 17 | 9 | |
| Legumes (Serving/day) | 0 | 11 | 12 | 0.001 |
| | 1 | 22 | 9 | |
| | 2 | 34 | 15 | |
| | 3 | 22 | 3 | |
| | 4 | 8 | 9 | |
| | <5 | 2 | 51 | |
| Whole Grains (Serving/day) | 0 | 34 | 40 | 0.803 |
| | 1 | 15 | 9 | |
| | 2 | 15 | 10 | |
| | 3 | 9 | 11 | |
| | 4 | 7 | 5 | |
| | 5 | 19 | 24 | |
| Artificial Sweetener | Yes | 34 | 36 | 0.408 |
| | No | 65 | 69 | |
| Caffeine Sources | Tea | 38 | 36 | 0.843 |
| | Coffee | 53 | 54 | |
| | Energy Drinks | 8 | 8 | |
| | Nothing | 0 | 0 | |
| Caffeine (cup/day) | 0 | 10 | 8 | 0.200 |
| | 1-2 | 11 | 12 | |
| | 3-4 | 6 | 1 | |
| | 4-5 | 2 | 1 | |

P value was completed using Paired-Samples T Test to compare the results of pre- and during Ramadan.

These findings may be a result of an increase in fat and sugar intake during RIF, which is compatible with our results that indicate an increase in pastries and dessert consumption in addition to a decrease in the percentage of fat after Ramadan. Also, Madkour et al. (2022) indicated high consumption of sugar after Ramadan, which increased from the mean of 65.9 g/d to 107.7 g/d [32]. That is consistent with our results, which showed a significant increase in the daily frequency of dessert consumption. Results of the present study showed a significant decrease in consumption of red meat, which may indicate lower overall animal protein intake as a result of a decreased number of meals to 1 meal a day. Comparable to this result, Madkour et al. (2022) showed a significant decline in protein intake from 108.3 g/day to 90 g/day after RIF days [32].

The RIF significantly alters dietary patterns among healthy adults, as it involves abstaining from food and drink from dawn to sunset for an entire month [36]. During Ramadan, individuals typically consume two main meals: Suhoor (pre-dawn meal) and Iftar (meal to break the fast at sunset). This shift in meal timing and frequency leads to notable changes in dietary intake compared to non-fasting periods. Studies have shown that after RIF, there is an increase in the consumption of high-calorie foods, sweets, and fried items, particularly after Iftar [36]. Conversely, the intake of fruits, vegetables,

and water may decrease due to the limited eating time [36]. These dietary changes can impact overall nutrient intake, with potential reductions in fiber, vitamins, and minerals [29]. Comparing these patterns to year-round dietary habits, it is evident that RIF introduces unique challenges and opportunities for maintaining a balanced diet [37]. Furthermore, there is a notable but insignificant decrease in physical activity after Ramadan, which is presented as a sum of MetS [38]. The decline in physical activity may be explained by the fear of thirst and dehydration, especially because of the length and heat of Ramadan days.

Changes in mental stress after RIF are widely varied between studies. Our results revealed that there was a significant decrease in mental stress among the participants after Ramadan. This result contradicts the result revealed by Al-otaibi et al. (2023), which indicated an increase in mental stress score [6]. On the other hand, Boukhris et al. (2019) demonstrated that mental stress remained unchanged after Ramadan [37]. Results of the current study indicated that sleep hours decreased significantly during Ramadan. This result is consistent with another study that was conducted by Bener et al. (2021), which revealed that there were fewer sleep hours during Ramadan [39]. In contrast, Margolis and Reedet (2004) showed that there was an increase in daytime sleep during Ramadan [40]. However Majid et al. (2023) demonstrated that there is no change in total sleep time between before and after Ramadan [41]. It is worthy note that changes in sleep time might contribute to the changes in serum levels of leptin, insulin, and cortisol; these factors could affect daily energy consumption and indirectly explain some of the body weight variation during and after RIF [42].

Reductions in oxidative stress and inflammatory markers can explain changes in mental stress during RIF through several mechanisms [10]. The RIF can decrease oxidative stress by enhancing the body's natural antioxidant defenses, which improves brain function and reduces anxiety and depression [43]. Additionally, fasting lowers levels of inflammatory markers like C-reactive protein and interleukin-6, which are linked to mental health issues [44]. By reducing inflammation, fasting helps alleviate symptoms of depression and anxiety [45]. Fasting also influences stress hormones like cortisol, leading to reduced stress and anxiety [46]. These hormones effects contribute to the positive changes in mental stress observed during RIF.

Duration of RIF vary from 12 to 20 hours based on location of countries and time of the year, affecting physiological and psychological responses of population [47]. Longer fasting time can alter metabolic responses, hydration status, and energy balance, contributing to variations of studies [13]. Additionally, cultural dietary habits during RIF differ, with some regions consuming more sweets and fried foods, while others focus on balanced meals [14]. These dietary differences impact body weight changes, metabolic health, and stress levels, leading to inconsistence of study outcomes.

The current study has several limitations, firstly, university students, in particular, have unique dietary, and physical activity patterns that differ from those of the general population. This factors could impact the generalizability of the findings. Furthermore, the study's sample size was relatively small, which could be expanded in future research to enhance the robustness of the findings. Finally, we did not calculate the daily calorie intake, which is an important factor to consider in discussing weight loss. A longitudinal study following participants over a longer period would help assess the long-term effects of RIF on health outcomes.

## 5. Conclusion

Fasting positively impacts lifestyle parameters, body weight, and stress levels among university students. Significant improvements were observed in dietary habits and sleep duration, contributing to overall well-being. Anthropometric indices, including reductions in body weight, BMI, and BRI, showed notable improvements. Additionally, stress levels decreased significantly during RIF, highlighting its potential benefits for mental health. These findings suggest that RIF can be an effective strategy for enhancing both physical and psychological health.

## Acknowledgments

The authors thank the participants for their patience and great help.

## Author contributions

**Conceptualization:** Anfal AL-Dalaeen.

**Data curation:** Anfal AL-Dalaeen.

**Formal analysis:** Anfal AL-Dalaeen, Eman Saleh, Sally Atawneh.

**Funding acquisition:** Anfal AL-Dalaeen.

**Investigation:** Anfal AL-Dalaeen.

**Methodology:** Anfal AL-Dalaeen, Eman Alhasan, Eman Saleh, Sally Atawneh, Tala Barakat.

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
