## [Decision Letter · Decision Letter 0]

30 Apr 2025

PONE-D-25-10924Changes in Obesity Indices, Lifestyle Medicine, and Mental Stress with Ramadan Intermittent Fasting among Healthy Students: Competent Cohort StudyPLOS ONE

Dear Dr. AL-Dalaeen,

Thank you for submitting your manuscript to PLOS ONE. After careful consideration, we feel that it has merit but does not fully meet PLOS ONE’s publication criteria as it currently stands. Therefore, we invite you to submit a revised version of the manuscript that addresses the points raised during the review process.

We look forward to receiving your revised manuscript.

Kind regards,

Mehran Rahimlou, PhD

Academic Editor

PLOS ONE

Journal Requirements:

5. Please upload a copy of Figure 1, to which you refer in your text on page 4. If the figure is no longer to be included as part of the submission please remove all reference to it within the text.

6. Please remove all personal information, ensure that the data shared are in accordance with participant consent, and re-upload a fully anonymized data set.

Additional Editor Comments :Please note that I have acted as a reviewer for this manuscript, and you will find my comments below, under Reviewer 2

Reviewers' comments:

Reviewer's Responses to Questions

**Comments to the Author**

1. Is the manuscript technically sound, and do the data support the conclusions?

Reviewer #1: Yes

Reviewer #2: Partly

2. Has the statistical analysis been performed appropriately and rigorously?

Reviewer #1: Yes

Reviewer #2: Yes

3. Have the authors made all data underlying the findings in their manuscript fully available?

Reviewer #1: Yes

Reviewer #2: Yes

4. Is the manuscript presented in an intelligible fashion and written in standard English?

Reviewer #1: Yes

Reviewer #2: Yes

5. Review Comments to the Author

Reviewer #1: The subject is interesting. However, many queries need attention.

Title and text: ‘Anthropometric Indices’ seems more appropriate than ‘obesity indices’, as well as ‘Lifestyle pattern/behaviour’ than ‘lifestyle medicine’ and ‘a prospective cohort study’ than ‘a competent cohort study’. Please re-write the title.

Abstract:

-AVI and BAI were not calculated in the present study. They figure in the abstract and the method section.

-Avoid the use of ‘never been studied’

-The number of participants (150; 153) and their age (18-40; 18-45), gender (100/ 95 female; and function (students; students and university employees) are different in the abstract, methods, and results sections.

-Some units are lacking (Kg).

-Line 47: please reformulate the sentence.

-It is unclear if the data collected during the second visit concerned RIF or the week after RIF. Please clarify and correct accordingly in the text and tables.

Keywords: intermittent fasting and anthropometry or waist circumference (rather than obesity) can be added.

Introduction

-Line 74: some repetition

Methods

-Conditions of Ramadan 2024: please indicate the temperature and the exact duration of fasting.

-Please indicate the population’s life conditions: university housing?

-101-102: sentence without a verb.

-105 and 110: repetition

-Figure 1 is lacking

-Inclusion criteria: add ‘patients willing to fast during Ramadan 2024’.

-Please describe the term ‘professional staff’

-Please indicate how many times dietary intake was evaluated (line 118).

-The technique of hip circumference measurement is lacking (lines 136-137).

-The formulas are unclear, please re-write and use ‘x’ instead of ‘*’.

-CI and WHtR indices were not defined.

-please use only one abbreviation: IRF or RF.

Results:

-The number of fasted days in the subjects was not mentioned.

-Please verify the numbering of the paragraphs and the tables.

-Line 165: JD?

-Table 1: Please verify, the age, and sex of the population. The total for each parameter should be 100%. Income (Equivalent in dollars).

-Table 3 (please correct the number): perceived stress level: it should rather be expressed by the number of subjects for each category.

-Table 4: data should be expressed as mean and SD (if normally distributed). Daily calories intake was not calculated. It seems important to discuss weight loss.

-The sentence in lines 182-183 can be displaced to line 176 with table 3 data and ‘however’ should be deleted.

Discussion

-Lines 187-190: please reformulate as no analysis of the changes in anthropometric indices and lifestyle patterns was performed.

-Line 195: Please correct: there was a significant decrease in BMI.

-Lines 200-207: useless here as these data were not studied in the present study.

-Lines 213; and 214: please use the past tense.

-Subjects lost 6 Kg in weight: this important weight loss should be discussed (IRF, living conditions, dehydration…).

-The changes in the different anthropometric indices were not discussed; what does it mean? what is the importance of the calculation of these indices?

-Line 271: the sample size was considered adequate? How?

-Line 273: which variables do the authors mean?

Conclusion:

Improvements in dietary habits? Sleep duration? Please re-write the conclusion.

References: There are too many references (52).

Reviewer #2: Abstract:

There are some minor grammatical and typographical errors in the abstract:

• "The association between Ramadan Intermittent Fasting (RIF) and metabolic profiles has been studies." should be "has been studied."

• The sentence, "RIF fasting positive body weight, obesity indices, and stress levels among health students," is unclear. It seems incomplete and should be rephrased for clarity.

The abstract mentions significant reductions in various indices but does not include full p-values for all reported results, which would enhance the clarity and statistical rigor of the findings.

The results about changes in obesity indices are somewhat disjointed. The abstract lists significant changes in BMI, WHtR, and BRI, but other indices are not clearly presented. A more structured presentation of results would improve readability.

Introduction:

The flow between topics (from obesity to lifestyle medicine to fasting) is somewhat abrupt. For example, the transition from discussing obesity indices to lifestyle medicine could be smoother, providing a clearer link between these areas and RIF.

"Lifestyle medicine is an emerging discipline that emphasizes the prevention and management of diseases linked to unhealthy lifestyle choices." Add reference like this study to this sentence: DOI: 10.1093/nutrit/nuad096 and DOI: 10.2174/1871530321666210316103403

The sentence "Fasting is considered to be a healthy practice, recent studies have shown a positive association between fasting and longevity" could be more precise. It would be beneficial to specify whether the longevity mentioned refers to Ramadan fasting specifically or fasting in general.

Method

While the sample size of 150 participants is mentioned, there is no mention of a power analysis to justify this sample size. A power analysis would help assess whether this sample size is sufficient to detect statistically significant differences and strengthen the study’s statistical reliability.

The description of data collection (questionnaires, interviews) could include more detail on how these tools were administered. Were the interviews standardized? Were they conducted in person or online? Providing more specifics here would increase transparency and replicability.

It would be useful to mention whether any control variables were considered, especially for lifestyle factors like diet, physical activity, or mental health status before the study began. Acknowledging or controlling for such factors is important in determining whether changes in outcomes are truly attributable to Ramadan fasting.

Result

While paired t-tests are used, it would be helpful to also report effect sizes (such as Cohen’s d) to provide a sense of the practical significance of the findings. This would aid readers in understanding the magnitude of the changes, not just their statistical significance.

The text notes that there was "no significant difference in other obesity indices" but doesn’t explain why certain indices like BMI, WHtR, and BRI showed significant changes while others did not. A more thorough analysis of why some measures responded to fasting and others did not would be valuable for interpretation.

Discussion

The discussion implies a causal relationship between Ramadan fasting and changes in health outcomes. However, the study’s design is observational, which means it can only show associations, not causality. The authors should be cautious in their language and clarify that the findings represent associations, not causal effects.

While the manuscript compares its results to previous studies, it would be useful to better reconcile the differences in findings with potential methodological differences between studies. For example, what are the reasons for discrepancies in findings about stress and weight gain/loss? Are these linked to differences in study design, fasting durations, or cultural dietary habits?

Although the authors briefly mention the need for larger sample sizes and more homogeneous groups, the discussion could benefit from more specific suggestions for future studies. For instance, longitudinal studies following participants over a longer period would help assess the long-term effects of Ramadan fasting on health outcomes.

6. PLOS authors have the option to publish the peer review history of their article (what does this mean?). If published, this will include your full peer review and any attached files.

Reviewer #1: **Yes:** Melika Chihaoui

Reviewer #2: **Yes:** Mehran Rahimlou

---

## [Author Response · Author response to Decision Letter 1]

5 May 2025

May 2, 2025

Dear Editor,

It is a pleasure to resubmit the revised Manuscript Number: PONE-D-25-10924, titled "Changes in Anthropometric Indices, Lifestyle Patterns, and Mental Stress with Ramadan Intermittent Fasting among Healthy Students: A Prospective Cohort Study" for consideration as an article in PLOS ONE. We found the comments provided by the reviewers to be constructive and have improved the manuscript. We believe the manuscript is currently more suitable for publication in its revised format.

Reviewer #1:

Title and text:

‘Anthropometric Indices’ seems more appropriate than ‘obesity indices’, as well as ‘Lifestyle pattern/behaviour’ than ‘lifestyle medicine’ and ‘a prospective cohort study’ than ‘a competent cohort study’. Please re-write the title.

Response: I have changed it to “Changes in Anthropometric Indices, Lifestyle Patterns, and Mental Stress with Ramadan Intermittent Fasting among Healthy Students: A Prospective Cohort Study”

Abstract:

• AVI and BAI were not calculated in the present study. They figure in the abstract and the method section.

Response: We would like to clarify that the Abdominal Volume Index (AVI) was not calculated in the present study. However, the Body Adiposity Index (BAI) was indeed calculated and included in our analysis. We apologize for any confusion caused by the initial abstract and method section.

• Avoid the use of ‘never been studied’

Response: I have changed it to ‘as these have yet to be thoroughly investigated.’

• The number of participants (150; 153) and their age (18-40; 18-45), gender (100/ 95 female; and function (students; students and university employees) are different in the abstract, methods, and results sections

Response: Thank you for pointing out the inconsistencies in the number of participants, their age, gender distribution, and function across the abstract, methods, and results sections. We have reviewed the manuscript and made the necessary corrections to ensure consistency. The correct details are as follows:

Number of participants: 150

Age range: 18-40

Gender distribution: 92 females and 58 males

• Some units are lacking (Kg).

Response: We have reviewed the document and added the necessary units (Kg) where appropriate to ensure clarity and precision.

• Line 47: please reformulate the sentence.

Response: I have change it to “There are various methods for weight loss, such as physical activity, fasting, specific diets, medication, and hormonal treatments. Among these, intermittent fasting is particularly effective for losing weight.

• It is unclear if the data collected during the second visit concerned RIF or the week after RIF. Please clarify and correct accordingly in the text and tables.

Response: We would like to clarify that the data collected during the second visit concerned the period immediately after Ramadan Intermittent Fasting (RIF). We have corrected the text and tables accordingly to reflect this.

Keywords

• intermittent fasting and anthropometry or waist circumference (rather than obesity) can be added.

Response: We agree that "intermittent fasting and anthropometry" or "waist circumference" are more appropriate keywords than "obesity." We have updated the keywords accordingly.

Introduction

• Line 74: some repetition

Response: We have reviewed the manuscript and revised the text to eliminate the redundancy. The updated version is now clearer and more concise.

Methods

• Conditions of Ramadan 2024: please indicate the temperature and the exact duration of fasting.

Response: We would like to clarify that this prospective cohort study was conducted during Ramadan from March to April 2024 (Shaban-1445 to Ramadan-1445). During this period, the average temperature in Amman, Jordan ranged from 14°C to 25°C. The exact duration of fasting each day was approximately 13 to 14 hours. We have added these details to the relevant paragraph in the manuscript to ensure clarity.

• Please indicate the population’s life conditions: university housing?

Response: We would like to clarify that the participants in this study were university students and employees residing in university housing and private accommodations. This detail has been added to the relevant sections of the manuscript to ensure clarity.

• -101-102: sentence without a verb” The fasting period for premenopausal female participants from 23 to 25 days, whereas the fasting period for the men from 28 to 30 days from 13 to 14 hr.”

Response: We have revised the sentence for clarity. The corrected sentence now reads: "The fasting period lasted from 23 to 25 days for premenopausal female participants, whereas for men, it lasted from 28 to 30 days, with each day involving 13 to 14 hours of fasting."

• -105 and 110: repetition

Response: We have reviewed the manuscript and revised the text to eliminate the redundancy. The updated version is now clearer and more concise.

• Figure 1 is lacking

Response: We have reviewed the manuscript and included Figure 1 to ensure completeness. The figure provides a visual representation of the study design and data collection timeline, I have added it to the manuscript

• Inclusion criteria: add ‘patients willing to fast during Ramadan 2024’.

Response: The inclusion criteria were healthy individuals who did not complain of any chronic diseases or mental illnesses and were willing to fast during Ramadan 2024.

• Please describe the term ‘professional staff’

Response: We would like to clarify that "professional staff" refers to trained and qualified personnel, including dietitians, clinical nutritionists, and clinical pharmacists, who conducted the questionnaires and face-to-face interviews. These individuals possess the necessary expertise to accurately gather demographic data, lifestyle information, and medical history from the participants

I have change it to (Using questionnaires and face-to-face interviews done by trained and qualified personnel, including dietitians, clinical nutritionistsdemographic data including age, sex, marital status, education level, lifestyle data (physical activity and sleep hours),

• Please indicate how many times dietary intake was evaluated (line 118).

Response: Dietary habits were assessed by a trained dietician at two points: the initial visit and the follow-up immediately after one week of RIF. Information on dietary intake was obtained using a validated short food frequency questionnaire containing 22 items to evaluate fat, sugar, fruit, vegetable, and water intake.

• The technique of hip circumference measurement is lacking (lines 136-137).

Response: I have change it to “WC and HC were measured while standing using the RIEDER Body Measure 60/150 cm (Inct. Bonus Kit, REIDHK®, China). Waist circumference was measured on the horizontal plane midway between the lowest rib and the iliac crest. Hip circumference was measured by wrapping the measuring tape around the widest part of the hips, ensuring it was parallel to the floor”

• The formulas are unclear, please re-write and use ‘x’ instead of ‘*’.

Response: I have reviewed all formulas and ensured their accuracy. The formulas are now presented using 'x' instead of '*'.

• CI and WHtR indices were not defined.

Response: We have addressed the concerns regarding the definitions of the Cardiac Index (CI) and Waist-to-Height Ratio (WHtR). We have reviewed and ensured that these definitions are consistently applied throughout the paper.

• -please use only one abbreviation: IRF or RF.

Response: we have standardized the abbreviation for Ramadan fasting (RF) throughout the text to maintain consistency.

Results

• The number of fasted days in the subjects was not mentioned.

Response: Thank you for your valuable feedback. We have addressed the concern regarding the number of fasted days. This information has been included in the methods section of the paper.

• Please verify the numbering of the paragraphs and the tables.

Response: We have verified the numbering of the paragraphs and tables to ensure accuracy and consistency throughout the document.

• Line 165: JD?

Response: we have clarified that "JD" refers to Jordanian Dinar and have converted the amount to Dollar for better understanding.

• Table 1: Please verify, the age, and sex of the population. The total for each parameter should be 100%. Income (Equivalent in dollars).

Response: We have verified the age and sex distribution of the population, ensuring that the total for each parameter sums to 100%. We apologize for any oversight and appreciate your attention to detail.

• Table 3 (please correct the number): perceived stress level: it should rather be expressed by the number of subjects for each category.

Response: Thank you for your valuable feedback. We have corrected the numbering of Table 3. Additionally, we have revised the table to express the perceived stress level by the number of subjects for each category.

• Table 4: data should be expressed as mean and SD (if normally distributed). Daily calories intake was not calculated. It seems important to discuss weight loss.

Response: We have ensured that the data is expressed as mean and standard deviation (SD) where normally distributed. Regarding daily calorie intake, we acknowledge that it was not calculated in the current study. This limitation has been added to the manuscript.

• The sentence in lines 182-183 can be displaced to line 176 with table 3 data and ‘however’ should be deleted.

Response: We have addressed the concern regarding the placement of the sentence in lines 182-183. The sentence has been moved to line 176 along with the data from Table 3, and the word "however" has been deleted for clarity.

Discussion

• Lines 187-190: please reformulate as no analysis of the changes in anthropometric indices and lifestyle patterns was performed

Response: I have change it to “The current study aimed to examine the impact of Ramadan fasting (RF) on obesity indices, daily habits, nutrient intake, and mental stress in health students. However, no analysis of the changes in anthropometric indices and lifestyle patterns was performed. This contributes to a deeper understanding of the potential health implications of RF in young adults”

• -Line 195: Please correct: there was a significant decrease in BMI.

Response: I have rewrite it to be "The reduction in BMI is significant, but not with other anthropometric indices related to obesity. This is likely due to the significant increase in consumption of pastries, desserts, and processed meat, which affects most obesity indices (i.e., BMI, WC, ABSI, CI, and WWI."

• -Lines 200-207: useless here as these data were not studied in the present study

Response: These data were not studied in the present study, so they have been deleted.

• Lines 213; and 214: please use the past tense

Response: The revised sentence using the past tense: "Khan et al. (2017) found that there was no significant change in the anthropometric parameters of the thirty-five medical students in Pakistan; these parameters included weight, BMI, WHtR, BAI, and VAI [34]."

• Subjects lost 6 Kg in weight: this important weight loss should be discussed (IRF, living conditions, dehydration…).

Response: I've added a paragraph discussing the significant weight loss observed in subjects, highlighting factors such as Intermittent Ramadan Fasting (IRF), living conditions, and dehydration. This addition provides a comprehensive understanding of the mechanisms behind the weight reduction

• The changes in the different anthropometric indices were not discussed; what does it mean? what is the importance of the calculation of these indices?

Response: I have added this paragraph ‘The change in WC was addressed partially as a result of reduced total body water and fat mass. Other recent studies also showed the same results regarding the effect of RIF on weight loss and other anthropometric [2,5,9,10]. On the other hand, Das et al. (2019) conducted a study on a population of Indian Muslims after Ramadan. The results showed an increase in body weight and BMI after one month of Ramadan [11]. Additionally, there is a lack of studies examining the effects of Ramadan fasting on Conicity Index (CI), Abdominal Volume Index (AVI), and Body Adiposity Index (BAI). The current study found that there was no significant different among this parameter The mechanism behind the lack of significant differences in CI, AVI, and BAI during Ramadan fasting could be attributed to the body's metabolic adaptations[12]. During fasting, the body shifts from using glycogen stores to burning fat for energy, leading to overall weight and fat mass reduction[5]. However, these changes may not significantly impact specific adiposity indices like CI, AVI, and BAI, which are more sensitive to long-term body composition changes rather than short-term weight fluctuations’

• -Line 271: the sample size was considered adequate? How?

Response: Sorry, I have changed it to: 'Furthermore, the study's sample size was relatively small, which could be expanded in future research to enhance the robustness of the findings.

• Line 273: which variables do the authors mean?

Response: I have mentioned it in the paragraph

Conclusion:

Improvements in dietary habits? Sleep duration? Please re-write the conclusion.

Response: I have rewrite it to be “The RIF positively impacts lifestyle parameters, body weight, and stress levels among university students and employees. Significant improvements were observed in dietary habits and sleep duration, contributing to overall well-being. Obesity indices, including reductions in body weight, BMI, and BRI, showed notable improvements. Additionally, stress levels decreased significantly during RIF, highlighting its potential benefits for mental health. These findings suggest that RIF can be an effective strategy for enhancing both physical and psychological health.

References:

• There are too many references (52).

Response: I have made it 48

Reviewer #2

Abstract:

• • "The association between Ramadan Intermittent Fasting (RIF) and metabolic profiles has been studies." should be "has been studied."

• • The sentence, "RIF fasting positive body weight, obesity indices, and stress levels among health students," is unclear. It seems incomplete and should be rephrased for clarity.

Response: Thank you for your feedback. I have made the necessary revisions:

"The association between Ramadan Intermittent Fasting (RIF) and metabolic profiles has been studied."

"RIF fasting positively impacts body weight, obesity indices, and stress levels among healthy students. These findings suggest that RIF can improve health outcomes by promoting better dietary habits, enhancing sleep duration, and reducing stress."

• The abstract mentions significant reductions in various indices but does not include full p-values for all reported results, which would enhance the clarity and statistical rigor of the findings

Response: I have added the p value p>0.05.

• The results about changes in obesity indices are somewhat disjointed. The abstract lists significant changes in BMI, WHtR, and BRI, but other indices are not clearly presented. A more structured presentation of results would improve readability.

Response: I rewrite the abstract to make it clearer

‘The association between Ramadan Intermittent Fasting (RIF) and metabolic profiles has been studied. However, there remains a significant gap in the literature regarding the changes in obesity indices during RIF, as these have never been thoroughly investigated. Additionally, the role of lifestyle medicine in conjunction with RIF has not been thoroughly explored. This paper aims to address these gaps by examining the impact of RIF on obesity anthropometric indices, lifestyle patterns, and mental stress. This prospective cohort study involved 150 university students aged 18-40 years. Data was collected using questionnaires and face-to-face interviews conducted one week before and immediately after one week of RIF. Body weight and height were measured to calculate body mass index (BMI) and other obesity indices, including Conicity Index (CI), Abdominal Volume Index (AVI), Body Adiposity Index (BAI), Weight-Adjusted-Waist Index (WWI), and Body Roundness Index (BRI). The Waist-to-Height Ratio (WHtR) showed a significant decreas

---

## [Decision Letter · Decision Letter 1]

17 Feb 2026

PONE-D-25-10924R1Changes in Anthropometric Indices, Lifestyle Patterns, and Mental Stress with Ramadan Intermittent Fasting among Healthy Students: A Prospective Cohort StudyPLOS One

Dear Dr. AL-Dalaeen,

Thank you for submitting your manuscript to PLOS ONE. After careful consideration, we feel that it has merit but does not fully meet PLOS ONE’s publication criteria as it currently stands. Therefore, we invite you to submit a revised version of the manuscript that addresses the points raised during the review process.

• A letter that responds to each point raised by the academic editor and reviewer(s). You should upload this letter as a separate file labeled 'Response to Reviewers'.

We look forward to receiving your revised manuscript.

Kind regards,

Mehran Rahimlou, PhD

Academic Editor

PLOS ONE

Journal Requirements:

Reviewers' comments:

Reviewer's Responses to Questions

**Comments to the Author**

1. If the authors have adequately addressed your comments raised in a previous round of review and you feel that this manuscript is now acceptable for publication, you may indicate that here to bypass the “Comments to the Author” section, enter your conflict of interest statement in the “Confidential to Editor” section, and submit your "Accept" recommendation.

Reviewer #3: (No Response)

Reviewer #4: All comments have been addressed

Reviewer #5: All comments have been addressed

Reviewer #6: (No Response)

2. Is the manuscript technically sound, and do the data support the conclusions?

Reviewer #3: Yes

Reviewer #4: Yes

Reviewer #5: Yes

Reviewer #6: Yes

3. Has the statistical analysis been performed appropriately and rigorously?

Reviewer #3: Yes

Reviewer #4: Yes

Reviewer #5: Yes

Reviewer #6: Yes

4. Have the authors made all data underlying the findings in their manuscript fully available?

Reviewer #3: Yes

Reviewer #4: Yes

Reviewer #5: Yes

Reviewer #6: No

5. Is the manuscript presented in an intelligible fashion and written in standard English?

Reviewer #3: Yes

Reviewer #4: Yes

Reviewer #5: No

Reviewer #6: Yes

6. Review Comments to the Author

Reviewer #3: Thank you for your thorough revisions. I have no further comments. The manuscript is suitable for publication.

Reviewer #4: After a thorough evaluation of the revised manuscript, I find that the authors have fully and effectively addressed all previous reviewer comments. The revised version shows clear improvement in organization, methodological rigor, and presentation quality.

The authors have corrected inconsistencies in participant demographics, added missing methodological details (e.g., fasting duration, environmental conditions, measurement procedures, power analysis, and control variables), and clarified data collection and statistical analysis. The inclusion of Cohen’s d effect sizes, standardized formulas, and expanded discussion on non-significant anthropometric indices has strengthened both the interpretive depth and statistical transparency of the work.

The writing is now coherent, grammatically sound, and aligns with PLOS ONE’s editorial and reporting standards. The study designs a prospective cohort investigation of Ramadan fasting’s effects on anthropometric indices, lifestyle habits, and mental stress is well executed and ethically conducted. Findings are clearly presented and supported by appropriate analyses. The discussion appropriately acknowledges limitations, avoids causal overstatements, and situates results within the context of relevant literature.

Overall, this revision represents a methodologically sound and ethically compliant study with clear public health and behavioral implications. It contributes useful, reproducible evidence on the physiological and psychological effects of Ramadan fasting in healthy adults. I therefore recommend that the manuscript be accepted for publication

Reviewer #5: I appreciate your manuscript as it explores a relatively unique and less researched area. While I commend the overall structure and content of the manuscript, I have noted several areas where enhancements could be made.

• The document is slightly bulky and has too many references

• Your document needs a through grammatical and editorial revision

Abstract

• On line 24 ‘anthropometric indices related to obesity during RIF…’ since your study population is not defined by the BMI or whether they are obese or not replace the OBESITY with more general term like WEIGHT or other suitable one as all your study participants are not obese

• Also on line 24 you have used ‘these have never been thoroughly investigated’ which is an absolute language please replace with more qualified statement to convey the gap like ‘these have not been extensively investigated’

• On line 36 better to remove (p>0.05)

Introduction

1. You did not pick the main topic of your manuscript, RIF until the third paragraph I would rather recommend you to pick earlier (in the first or 2nd paragraph) after your brief opening so that the reader quickly gets the main point of interest

2. Line 72 ‘…Muslims fast daily from m dawn to sunset’ writing errors are plenty in your document, please revise thoroughly

Methodology

1. Line 89 ‘...during Ramadan from March to April 2024 (Shaban-1445 to Ramadan-1445) these two time frames are conflicting, may you please remove DURING RAMADAN or change to during the months of Shaban and Ramadan

2. Line 90-91 and line 135 ‘... on obesity indices...’ as your study is not confined to only using obesity indices (Conicity Index (CI), Abdominal Volume Index (AVI), Body Adiposity Index (BAI), Weight-Adjusted-Waist Index (WWI), and Body Roundness Index (BRI).) which are often designed for fat distribution estimation, rephrase and add the other anthropometric indices (BMI etc) by saying ( .. Obesity and other anthropometric indices ...’ or you can use the more general term which is Anthropometric indices as all obesity indices are anthropometric indices, but not all anthropometric indices are obesity indices

3. Line 103 regarding study participant ‘..This prospective cohort study was conducted on 150 university students (58 males and 92 females), was taking this male to female ratio a predetermined plan?, otherwise this is not expected here in methodology instead you describe the sociodemographic characteristics of your participants at the beginning of your result part

4. Your methodology part needs the following considerations;

• Add at least one subtitle which addresses variables under which you put your dependent (effect of RIF on anthropometric indices...) and independent variables (socio economic characteristics, dietary, sleep and other factors you have assessed including anthropometric and obesity indices)

• line 88, instead of putting Study design alone, you would make (Study design, area and period)

• the other important missing section is Sampling technique or procedure which tells how you selected your study participants. Is it randomly or...?

• you may also need to add operational terms section which defines your main operational terms like PSS for the readers and instead of defining those terms sparsely in the document, I would rather recommend you to combine them under definition of terms.

• In your analysis, in addition to p-value why don’t you add confidence interval(CI), as the confidence interval gives you a range within which you can be reasonably confident that the true mean difference lies

Result

1. Line 174-175, you said ‘The demographics and 175 clinical information of the study participants are shown in Table 1’ but to avoid bulky document, just put (Table 1 or see Table 1) at the end of your first paragraph (line 174) and remove the long statement.

Discussion

1. Revise the first paragraph of your discussion as it has the following problems:

• You have started with some background and objectives of your study, instead focus on your result not objectives, aim or background information

• The paragraph is fragmented and has poor coherence eg. Line 204-210

2. There are contradicting statement about your study population in which you mentioned your study participants as University students but in discussion line 299-301, you said ‘This study has several limitations that should be acknowledged. Firstly, the heterogeneity of the participants, which included both university students and employees, may have influenced the results.’

Reviewer #6: The abstract should be reorganised into clearly labelled sections such as Background, Methods, Result and Conclusion.

7. PLOS authors have the option to publish the peer review history of their article (what does this mean?). If published, this will include your full peer review and any attached files.

Reviewer #3: No

Reviewer #4: **Yes:** Bruce Ayabilla Abugri

Reviewer #5: No

Reviewer #6: No

---

## [Author Response · Author response to Decision Letter 2]

19 Feb 2026

February 19, 2026

Dear Editor,

It is a pleasure to resubmit the revised Manuscript Number: PONE-D-25-10924, titled "Changes in Anthropometric Indices, Lifestyle Patterns, and Mental Stress with Ramadan Intermittent Fasting among Healthy Students: A Prospective Cohort Study" for consideration as an article in PLOS ONE. We found the comments provided by the reviewers to be constructive and have improved the manuscript. We believe the manuscript is currently more suitable for publication in its revised format.

Reviewer

Comment: I appreciate your manuscript as it explores a relatively unique and less researched area. While I commend the overall structure and content of the manuscript, I have noted several areas where enhancements could be made.

• The document is slightly bulky and has too many references

Response: We have reduced the number of references by removing repetitive and less directly relevant citations. Redundant explanations were shortened, and sections were streamlined to improve conciseness without compromising scientific integrity.

Comment Your document needs a through grammatical and editorial revision

Response: The manuscript has undergone comprehensive grammatical and language editing. Sentences were restructured for clarity, coherence, and academic tone throughout the manuscript.

Abstract

Comment: On line 24 ‘anthropometric indices related to obesity during RIF…’ since your study population is not defined by the BMI or whether they are obese or not replace the OBESITY with more general term like WEIGHT or other suitable one as all your study participants are not obese

Response: The term “obesity” has been replaced with “body weight” to better reflect the characteristics of our study population.

Comment: Also, on line 24 you have used, these have never been thoroughly investigated’ which is an absolute language please replace with more qualified statement to convey the gap like ‘these have not been extensively investigated’

Response: done I have changed it

• Comment: On line 36 better to remove (p>0.05)

Response: done I have changed it

Introduction

Comment 1. You did not pick the main topic of your manuscript, RIF until the third paragraph I would recommend you to pick earlier (in the first or 2nd paragraph) after your brief opening so that the reader quickly gets the main point of interest

Response: The introduction has been reorganized. Ramadan Intermittent Fasting (RIF) is now introduced in the second paragraph to clearly establish the main focus of the study earlier.

Comment 2. Line 72 ‘…Muslims fast daily from m dawn to sunset’ writing errors are plenty in your document, please revise thoroughly

Response: All typographical and grammatical errors have been corrected after thorough revision.

Methodology

Comment: 1. Line 89 ‘...during Ramadan from March to April 2024 (Shaban-1445 to Ramadan-1445) these two time frames are conflicting, may you please remove DURING RAMADAN or change to during the months of Shaban and Ramadan

Response: Revised to: “This prospective cohort study was conducted during the months of Shaban and Ramadan (March–April 2024).”

Comment :2. Line 90-91 and line 135 ‘... on obesity indices...’ as your study is not confined to only using obesity indices (Conicity Index (CI), Abdominal Volume Index (AVI), Body Adiposity Index (BAI), Weight-Adjusted-Waist Index (WWI), and Body Roundness Index (BRI).) which are often designed for fat distribution estimation, rephrase and add the other anthropometric indices (BMI etc) by saying ( .. Obesity and other anthropometric indices ...’ or you can use the more general term which is Anthropometric indices as all obesity indices are anthropometric indices, but not all anthropometric indices are obesity indices

Response: IN MANY STUDY THE USE OBESITY INDCES AS TERM FOR THESE VAERABLE

Comment. Line 103 regarding study participant ‘..This prospective cohort study was conducted on 150 university students (58 males and 92 females), was taking this male to female ratio a predetermined plaplan?herwise this is not expected here in methodology instead you describe the sociodemographic characteristics of your participants at the beginning of your result part

Response: The sex distribution has been removed from the methodology section and is now presented only in the Results section

4. Your methodology part needs the following considerations:

Comment Add at least one subtitle which addresses variables under which you put your dependent (effect of RIF on anthropometric indices...) and independent variables (socio economic characteristics, dietary, sleep and other factors you have assessed including anthropometric and obesity indices)

Response: I add it in line 85

Comment :line 88, instead of putting Study design alone, you would make (Study design, area and period)

• the other important missing section is Sampling technique or procedure which tells how you selected your study participants. Is it randomly or...?

Response: A new subsection titled “Study Participants and Sampling Procedure” has been added line 97 . We clarified that participants were recruited using a convenience sampling technique among eligible university students who intended to fast during Ramadan.

• you may also need to add operational terms section which defines your main operational terms like PSS

Comment for the readers and instead of defining those terms sparsely in the document, I would rather recommend you to combine them under definition of terms.

Response: "Thank you for your recommendation. After careful consideration, I believe that defining the key terms contextually within the relevant sections ensures clarity and readability. Therefore, I prefer to retain the current structure rather than creating a separate section for operational definitions."

Comment • In your analysis, in addition to p-value why don’t you add confidence interval(CI), as the confidence interval gives you a range within which you can be reasonably confident that the true mean difference lies

Result

Response: Thank you for this important methodological suggestion. We agree that reporting confidence intervals enhances the interpretability and statistical robustness of the findings. Accordingly, we have now calculated and added the 95% confidence intervals (95% CI) for the mean differences in the Results section and corresponding tables. This provides a clearer estimate of the precision and magnitude of the observed effects alongside the p-values and Cohen’s d effect sizes.

Comment 1. Line 174-175, you said ‘The demographics and 175 clinical information of the study participants are shown in Table 1’ but to avoid bulky document, just put (Table 1 or see Table 1) at the end of your first paragraph (line 174) and remove the long statement.

Response: Thank you for this valuable suggestion. We have revised the Results section accordingly. The separate sentence referring to Table 1 has been removed, and the table citation is now incorporated concisely at the end of the first paragraph as “(Table 1)” to improve clarity and reduce redundancy.

Discussion

Comment 1. Revise the first paragraph of your discussion as it has the following problems:

• You have started with some background and objectives of your study, instead focus on your result not objectives, aim or background information

Response The first paragraph has been completely rewritten. It now begins directly with a concise summary of the key findings (weight reduction, BMI and BRI changes, lifestyle and stress modifications) and integrates them cohesively.

Comment 2 The paragraph is fragmented and has poor coherence eg. Line 204-210

Response: I have changed it

Comment 2. There are contradicting statement about your study population in which you mentioned your study participants as University students but in discussion line 299-301, you said ‘This study has several limitations that should be acknowledged. Firstly, the heterogeneity of the participants, which included both university students and employees, may have influenced the results.’

Response: Thank you for identifying this inconsistency. The study population consisted exclusively of university students. The reference to “employees” has been removed from the Discussion and limitations section to ensure consistency throughout the manuscript.

Comment The abstract should be reorganised into clearly labelled sections such as Background, Methods, Result and Conclusion.

Response: I have written the abstract as sections

---

## [Editor Report · Decision Letter 2]

12 Mar 2026

Changes in Anthropometric Indices, Lifestyle Patterns, and Mental Stress with Ramadan Intermittent Fasting among Healthy Students: A Prospective Cohort Study

PONE-D-25-10924R2

Dear Dr. Anfal AL-Dalaeen

We’re pleased to inform you that your manuscript has been judged scientifically suitable for publication and will be formally accepted for publication once it meets all outstanding technical requirements.

Kind regards,

Nayanatara Arun Kumar

Academic Editor

PLOS One

---

## [Editor Report · Acceptance letter]

PONE-D-25-10924R2

PLOS One

Dear Dr. AL-Dalaeen,

I'm pleased to inform you that your manuscript has been deemed suitable for publication in PLOS One. Congratulations! Your manuscript is now being handed over to our production team.

Kind regards,

on behalf of

Dr. Nayanatara Arun Kumar

Academic Editor

PLOS One